# A Methodology to Design Static NCL Libraries

**Toi Le Thanh** [1,2,3] **, Lac Truong Tri** [1,2] **and Trang Hoang** [1,2,*]

1 Department of Electronics Engineering, Faculty of Electricals and Electronics Engineering, Ho Chi Minh City University of Technology (HCMUT), 268 Ly Thuong Kiet Street, District 10, Ho Chi Minh City 700000, Vietnam; lttoi.sdh19@hcmut.edu.vn or toilt@hufi.edu.vn (T.L.T.); ttlac.sdh20@hcmut.edu.vn (L.T.T.)

2 Vietnam National University Ho Chi Minh City, Linh Trung Ward, Thu Duc District, Ho Chi Minh City 700000, Vietnam

3 Department of Electronics Engineering, Faculty of Electricals and Electronics Engineering, Ho Chi Minh City University of Food Industry (HUFI), 140 Le Trong Tan Street, Tay Thanh Ward, Tan Phu District, Ho Chi Minh City 700000, Vietnam

* Correspondence: hoangtrang@hcmut.edu.vn

**Abstract:** The Null Convention Logic (NCL) based asynchronous design technique has interested researchers because this technique had overcome disadvantages of the synchronous technique, such as noise, glitches, clock skew and power. However, using the NCL-based asynchronous design method is difficult for university students and researchers because of the lack of standard NCL cell libraries. Therefore, in this paper, a novel flow is proposed to design NCL cell libraries. These libraries are used to synthesize NCL-based asynchronous designs. We chose the static NCL cell library to illustrate the proposed design solution because this library is one of the most basic NCL libraries. Static NCL cells in this library are designed based on the Process Design Kit 45nm technology and are implemented by the Virtuoso and the Design Compiler (DC) tool. In addition, the Ocean script and Electronic Design Automation (EDA) environment are used for supporting designs and simulations. A complete library of 27 NCL cells was designed to serve for study and research. We also implemented synthesis for NCL full adders using this library and compared our synthesis results with the results of other authors. The comparison results indicated that our results were a 20% improvement on power consumption.

**Keywords:** NCL cell library; threshold gate; asynchronous method; Null Convention Logic

## 1. Introduction

Synchronous circuits have played a significant role and have dominated the semiconductor industry [1]. This industry has continuously diminished the wire and transistor dimension. As a result, billions of transistors are integrated into a single chip, and low power and high-performance circuits will be created in the following technology generations. Furthermore, synchronous circuits use a clock signal to synchronize their operations. Therefore, the semiconductor industry must face clock-related issues, including clock skew, power consumption, noise, electromagnetic interference, and the complexity of clock network layout. These issues are considered future technological challenges for the semiconductor industry [2].

In contrast to synchronous circuit paradigms, asynchronous circuit paradigms synchronize their operations through the local handshake protocol. Therefore, asynchronous circuitry may eliminate the clock issues mentioned above [3]. Among asynchronous circuit paradigms, NCL is a quasi-delay-insensitive (QDI) logic paradigm used in commercial applications and is chosen to design asynchronous circuits [4]. Many studies of NCL-based asynchronous circuits are implemented, such as the complementary metal-oxide-semiconductor circuit design of threshold gates with latency [5], of which comparisons of NCL threshold gate models [3] and some relevant studies can be found in [6–12]. In most of the studies mentioned above, authors synthesized their designs in one of three approaches.

The first approach was to use tools to convert synchronous to asynchronous designs [13]. This approach makes it hard to optimize large-scale designs. In the second approach, the authors used a full-custom design flow to synthesize NCL-based designs. This flow is not suitable for complex designs. The last approach uses the conventional synchronous cell library to synthesize NCL-based designs [14]. This approach may prevent the NCL-based asynchronous designs from achieving optimal power. Therefore, the lack of NCL asynchronous cell libraries have caused many difficulties in research and development.

In state-of-the-art research, there were several flows suggested to design NCL asynchronous cell libraries [15,16]. These flows are complex, and authors used some of their own tools—a reason that causes difficulties for researchers who want to continue to inherit and develop. Therefore, we proposed a simple flow to design the NCL cell library using only the main commercial tools. Additionally, this flow also helps researchers themselves to easily create NCL cell libraries.

In this flow, cell schematic designs, cell symbol generation, and cell simulation to determine leakage power and input capacitance are implemented using Virtuoso. NCL static cells are designed based on PDK 45nm technology and are chosen as a case study. In addition, cell characterization is assisted by Ocean script to determine parameters, such as cell rise delay, cell fall delay, rise transition, fall transition, rise power and fall power. Thanks to the Ocean script, researchers saved time approaching a new method quickly.

The rest of this paper comprises three sections: Section 2 presents an overview of NCL, the proposed flow, and cell characterization. Section 3 provides results and discussions. Finally, Section 4 gives conclusions of our methodology to design the NCL cell library.

## 2. Materials and Methods

### 2.1. Null Convention Logic

NCL is an asynchronous logic and latency-insensitive model. To achieve delay insensitivity, NCL circuits must satisfy two rules: input-completeness and observability [17]. In terms of input-completeness, NCL designs require two following conditions: "The output may not transition from NULL to a complete set of DATA until the input value is purely DATA", and "The output may not transition from the DATA state to a NULL completer until the input value is completely NULL". About observability, this requires that there are no orphans transmitting through a gate. Orphans can be ignored through the isochronic fork assumption—wire delays are less than gate delays within a component [17]. The observability condition ensures that any gate transitions are observable at the output. To satisfy this observability condition, each transition occurring at each gate must transition at least one of the outputs.

The NCL-based circuit design method does not use a clock signal and is aimed at asynchronous circuits [5]. These asynchronous circuits always execute correctly, regardless of component and wire delays [18,19]. To achieve the delay target mentioned above, NCL circuits utilize dual-rail logic [18]. A conventional logic signal is generated by only one rail, while two rails form a dual-rail logic signal. Table 1 shows the conversion of a conventional logic signal to a dual-rail signal [20]. The value '11' is illegal because A0 and A1 rails are mutually exclusive.

**Table 1.** Dual-rail signal.

| Boolean Logic | Code | | |
|:---:|:---:|:---:|:---:|
| | **Dual-Rail Logic** | **A1** | **A0** |
| 0 | DATA0 | 0 | 1 |
| 1 | DATA1 | 1 | 0 |
| | NULL | 0 | 0 |
| | ILLEGAL | 1 | 1 |

Unlike conventional asynchronous circuits, NCL-based circuits use a set of twenty-seven threshold gates [18–20]. A general symbol of the thmn threshold gate is illustrated in Figure 1. Where n is the total number of inputs, m is the threshold value that means at least m of n inputs must become '1' state before the output becomes '1' state. Another type of threshold gate is denoted thnmWz1z2 . . . zm. It is a weighted threshold gate, where the input weights are z1, z2, . . . , and zm. For example, th23w2 is shown in Figure 1b, where the A input weight and the threshold value are two. Therefore, when the A input becomes a '1' state, the output will be asserted.

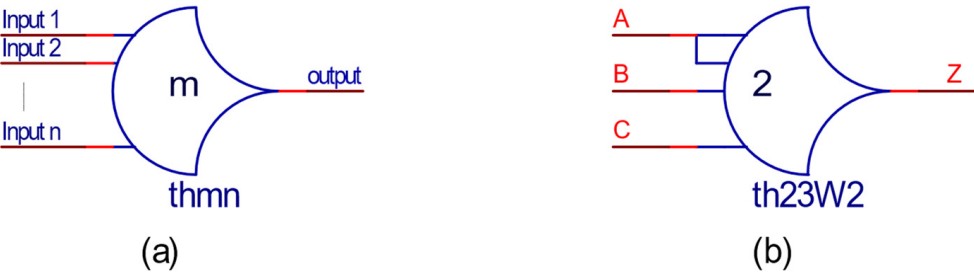

(a)                                                             (b)

**Figure 1.** The primary threshold gate (**a**) thmn; (**b**) Th23w2.

As presented above, NCL-based asynchronous circuits are designed by threshold gates. The general structure of a static CMOS threshold gate with hysteresis consists of five function blocks (set, reset, hold data, hold null and an inverter at the output), as shown in Figure 2. In this structure, the reset block and hold data block are complementary to each other and have standard structures which are depicted in Figure 3 [4,5]. The reset block is active when all inputs are in the '0' state, while the hold data block is active when at least one or more inputs are in the '1' state. Their structures only depend on the number of cell inputs. Therefore, threshold gates with the same number of inputs will have the same reset block and hold data block. Similarly, the set block and the hold null block complement each other, but their actual structures depend on the number of inputs and the threshold value.

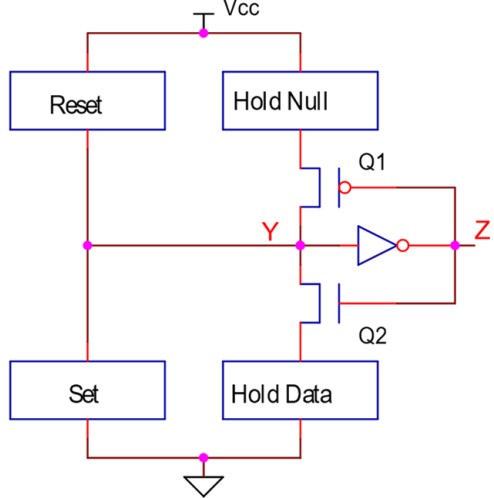

**Figure 2.** General structure of a static CMOS threshold gate.

Similar to the general structure of the static threshold gate, a general structure of the semi-static threshold gate comprises three function blocks (reset block, set block, and a weak feedback inverter at the output) [3,5]. In this structure, when both set and reset blocks are off, the logic level on node Y will be remained by this inverter. In addition, this weak inverter will be influenced by noise on node Y if it is too small.

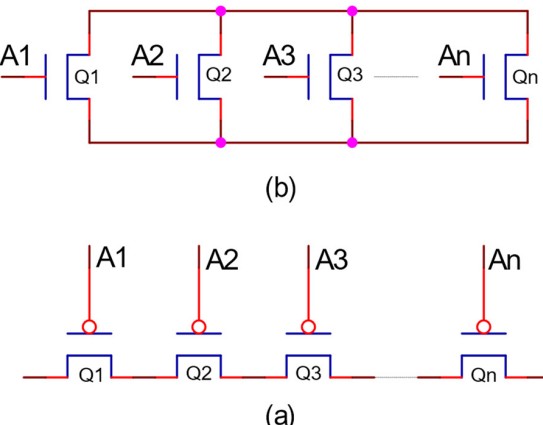

**Figure 3.** General structure of reset block and hold data block: (**a**) reset block; (**b**) hold data block.

In many applications related to real-time computing, such as signal processing, the flow of the input data is continuous at the minimum speed. In these cases, a feedback mechanism is not essential to maintain the state information. Therefore, the weak feedback inverter can be removed from the semi-static structure. As a result, a new paradigm is formed and is called dynamic threshold gates [3].

### 2.2. The Proposed Flow Chart to Design Standard NCL Cell Libraries

In this section, we present the proposed flow chart to design NCL cell libraries. This flow comprises ten steps depicted in Figure 4. Firstly, the cell specification analysis step is implemented to form the basis for the schematic circuit design step. The next step is to create the cell symbol to carry out the testbench circuit. This circuit is simulated to check the cell operations. If the cell operation test results are not good, we will go back to the schematic circuit design step. Otherwise, we will go to the simulation step at the corners. This step is implemented to measure leakage power and the input capacitance.

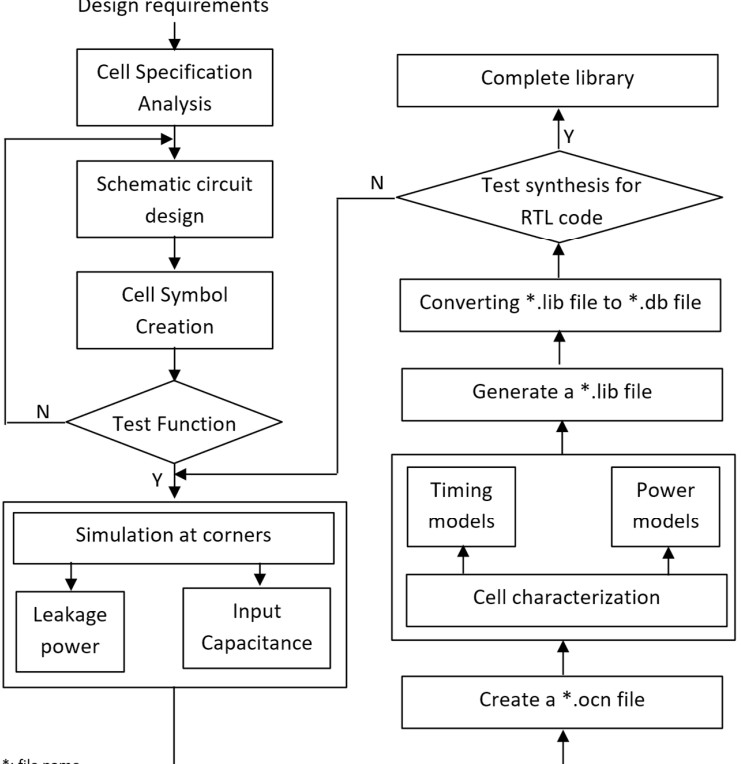

**Figure 4.** The proposed NCL cell library design flow chart.

Pin capacitance can be specified in all inputs and outputs. In most cases, it is only determined at input pins. Thus, the cell output capacitance is equal to zero [21]. The input capacitance value is computed by Equation (1), which represents the relationship of capacitance, voltage, and current.

$$I = C\frac{dV}{dt} \tag{1}$$

By providing pulse voltage to the input pin and measuring the current at the same point, we will calculate the input capacitance value by Equation (2):

$$C_{input} = \frac{\int_t^{t+\Delta t} I(t)dt}{\int_t^{t+\Delta t} dV} \tag{2}$$

where *I* is the current at an input pin, and it is created by charging and discharging the charge through the input capacitance.

Most normal cells only consume power when the output changes. However, other powers are dissipated as the cells are supplied with the voltage but are not active because the leakage current is not equal to zero. The sub-threshold current or the tunneling current through the gate oxide of MOS devices generates the leakage [21]. The leakage power is determined according to Equation (3):

$$P_{Leakage} = \sum I_{Leakage} V_{DD} \tag{3}$$

To determine the leakage power, we first list all input combinations of that cell and then compute the leakage power of every case by connecting the voltage supply line to the ground when inputs are low, or connecting the voltage supply line to $V_{DD}$ when inputs are high. The leakage power of a standard cell is equal to the average of all cases. Simultaneously, with the simulation step at corners, we perform the cell characterization to determine the timing and power models. As ADE does not have options or powerful commands to execute the repetitive tasks, the Ocean script is utilized to assist the cell characterization automatically. Cell characterization is represented in detail in Section 2.3 below. The parameters mentioned above, including leakage power, input capacitances, timing model, and power model, will contribute to forming the *.lib file [22]. This file complies with Synopsys standards. Subsequently, we use the Library Compiler tool of Synopsys to convert the *.lib file to the *.db file. To do this, read_lib and write_lib commands shall be used. "Read_lib <your path>/library.lib" command is used to read and compile the library file. If the compilation is successful, the program returns 1, and the *.db file is created by using "write_lib library_name -f db -o <your path>/library.db" command [23]. This *.db file is not only one of the crucial files in the library but also contains the essential parameters of cells and is used to synthesize NCL-based asynchronous circuits using the DC tool of Synopsys.

Finally, the synthesis step is carried out to check if the cell library works properly. In this step, we will write a piece of RTL code and synthesize it at the gate level. If the synthesis results are good (i.e., the design can be synthesized successfully) the NCL cell library will be completed. The complete library comprises 27 cells.

To illustrate the proposed flow, we choose any one of twenty-seven cells in the library, for instance, the th22 cell. As represented in Section 2.1, reset and hold data blocks are in standard forms and only depend on the total number of inputs. Therefore, the reset block is formed by two PMOS transistors in series, and the hold data block is formed by two NMOS transistors in parallel. To construct the schematic circuits of the set block and hold the null block, we conduct an analysis of their function. The set block is only active when both A and B input goes to the high level. For this condition, the switching expression of the set block also describes the function of this threshold gate, represented by Equation (4). The hold null block complements the set block, so the switching expression of this block is obtained by complementing Equation (4), and this result is shown in Equation (5). The

circuits to implement set and hold null blocks by using NMOS transistor networks and PMOS transistor networks are illustrated in Figure 5. The remaining steps will be continued in Section 2.3.

$$F(A, B) = AB \tag{4}$$

$$F(A, B) = \overline{A} + \overline{B} \tag{5}$$

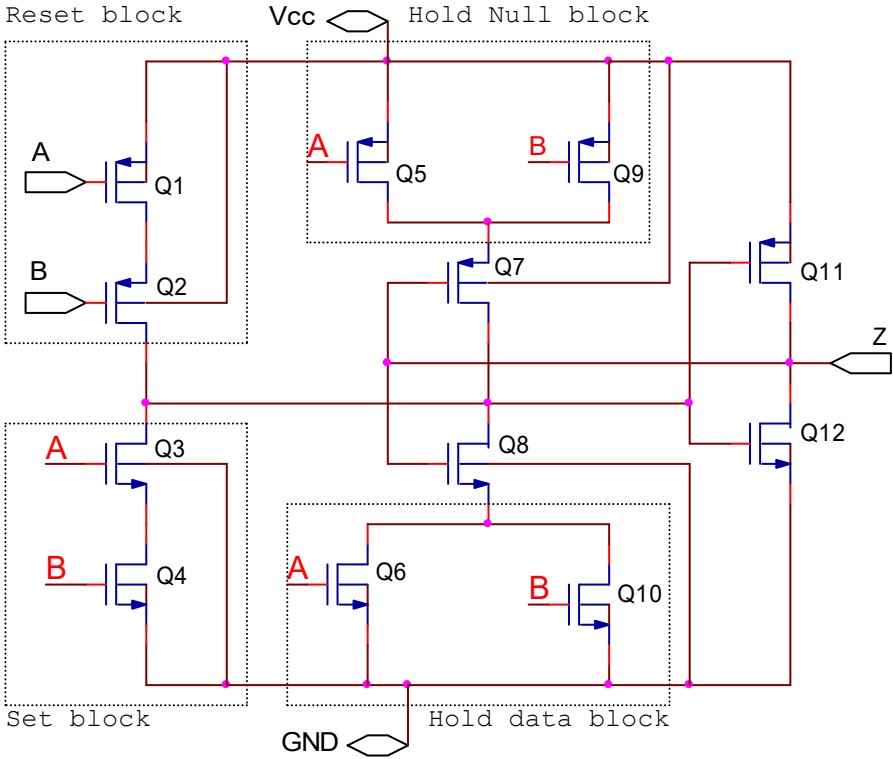

**Figure 5.** Threshold gate th22.

### 2.3. NCL Cell Characterization

Cell characterization is one of the most important steps in the flow because, in this step, cell timing models and power models are determined to form the library. We perform characterization for all cells, and the quantities of cell fall delay, cell rise delay, fall transition, rise transition, rise power, fall power, input capacitance and leakage power. In this section, 27 cells will use the same load capacitance $C_{load}$ (1.4 fF, 2.54 fF, 4.61 fF, 8.37 fF, 15.2 fF, 27.6 fF, 50.0 fF) and the same fall time and rise time of the input Vpulse waveform (0.01 ns, 0.0192 ns, 0.0368 ns, 0.0707 ns, 0.136 ns, 0.261 ns, 0.5 ns) to realize cell characterization. $C_{load}$ and slope values are determined based on Equation (6) and Elmore delay, respectively [24]. We measured $C_{load}$ and slope values with different drive strengths to get a range that cells can fall into.

$$C_{load} = (W_n \times L_n \times C_{ox}) + (W_p \times L_p \times C_{ox}) \tag{6}$$

where:

$C_{ox}$: Gate oxide capacitance;
$W_n$: width of the NMOS transistor;
$L_n$: length of the NMOS transistor;
$W_p$: width of the PMOS transistor;
$L_p$: length of the PMOS transistor.

To simulate all the cases, we must carry out the tasks manually because there are no options and powerful commands in the graphic user interface to carry out repetitive tasks, which is one of the greatest drawbacks of the ADE. In addition, there is no approach to characterize a normal cell automatically. Hence, in this subsection, Ocean language is

utilized to assist automatically implementing simulations within Cadence because it is one of the powerful script languages. The structure of the .ocn file includes three main parts: part 1 is to assign the $C_{load}$ and slope values to the two arrays, respectively; part 2 is to create loops for simulating 49 cases; and the last part is to measure rise transition, fall transition, cell rise delay, cell fall delay, rise power and fall power based on Figures 6–8 and Equation (7).

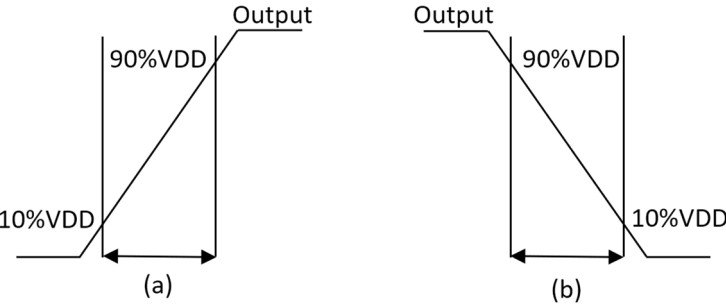

**Figure 6.** Transition time at output pin. (**a**) Rise transition. (**b**) Fall transition.

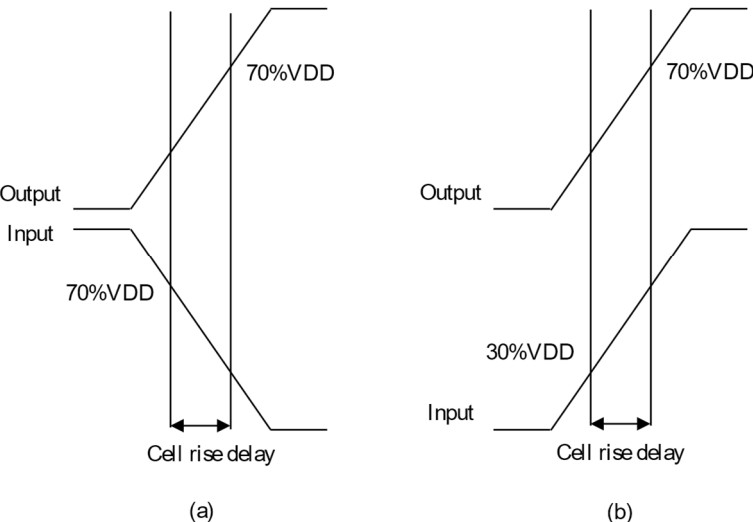

**Figure 7.** Cell rise delay. (**a**) Timing arc is negative unate. (**b**) Timing arc is positive unate.

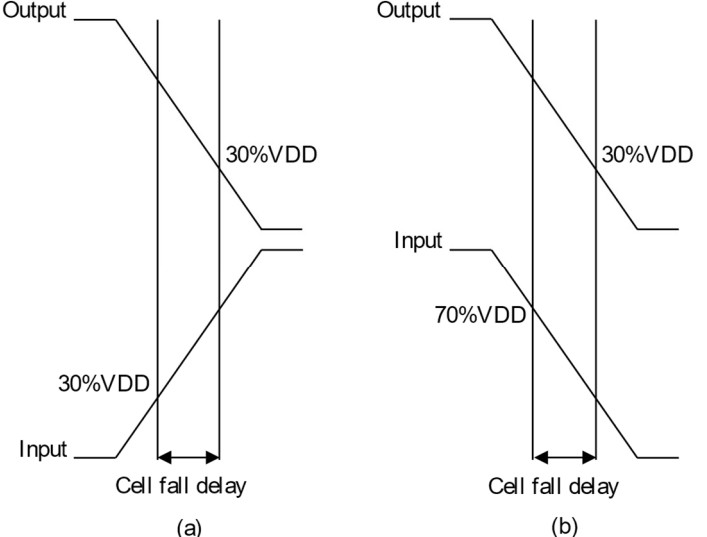

**Figure 8.** Cell fall delay. (**a**) Timing arc is negative unate. (**b**) Timing arc is positive unate.

The general structure of the .ocn file:
loadlist = list ("L$_1$" "L$_2$" "L$_3$" ... "L$_n$");
slopelist = list ("S$_1$" "S$_2$" "S$_3$" ... "S$_n$");
foreach (slopevar slopelist
foreach (loadvar loadlist
"Measure cell rise delay, cell fall delay, ... "
);
);
where:

L$_n$: load value

S$_n$: slope value

Besides the Ocean script, the calculator of Virtuoso is also used to perform cell characterization. The parameters mentioned above, such as fall time and rise time of the input voltage, and load capacitance must be determined clearly in the *.ocn file to run the simulation 49 times and calculate the timing and the dynamic power models, including fall transition, rise transition, cell rise delay, and cell fall delay. We do not use the Ocean script to assist in measuring the leakage power and the input capacitance because it is used to measure a range of values. The testbench circuit of the th22 gate is shown in Figure 9.

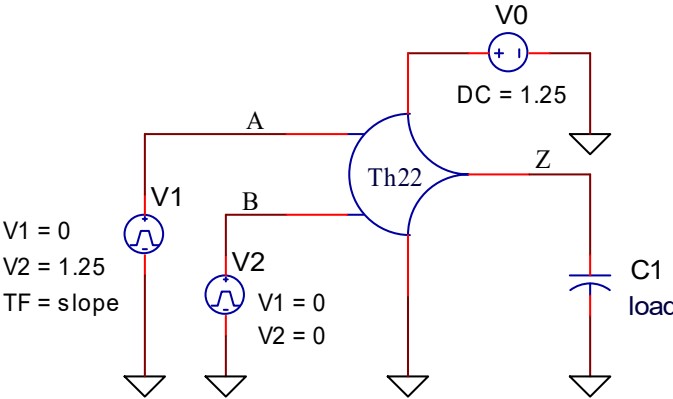

**Figure 9.** The testbench circuit.

The cell timing models are utilized to provide accurate timing for various cell cases in the design environment. Hence, non-linear delay paradigms are utilized to create a *.lib file because these paradigms are precise even if utilized for the submicron technology [21]. The timing models are computed for every timing arc of the cell. Delays and timing are table models. These paradigms must be determined clearly for all the cells in the library. The transition time at the output pin and the hysteresis via the cell for different combinations of the input transition time at the cell input and total output capacitance at the cell output pin is captured by the table models [21]. Figures 6–8 show calculating time values of the timing models (delay and transition time). The percentages (30%, 70%, 10%, 90%) are threshold values, which are specified clearly in the liberty file [25].

Dynamic power consists of rise power and fall power. Rise power is calculated in case the output changes from low to high. Similarly, fall power is calculated in case the output changes from high to low. The dynamic power is determined by Equation (7), where V0 is the power supply.

$$P_{dynamic} = \frac{V_{DD}}{T} * \int_{t}^{t+\Delta t} i_{("/V0/PLUS")}(t).dt \tag{7}$$

## 3. Results and Discussions

In this section, results of the processes in Section 2 are presented and discussed, including the cell function test, cell characterization and test synthesis for RTL code. In addition, we also make comparisons between our synthesis results and the results of other authors.

### 3.1. Function Test Results

In our proposed flow, it is necessary to check the function of the cell because if all possible combinations of the inputs are not fully checked, it can cause the results after performing cell characterization to be wrong. The simulation results, to test its function, are shown in Figures 10–12. Theoretically, when all inputs transition to low, the output will become low and when two inputs transition to high, the th22 gate output will become high. Figures (from Figures 10–12) indicate that the circuit works correctly.

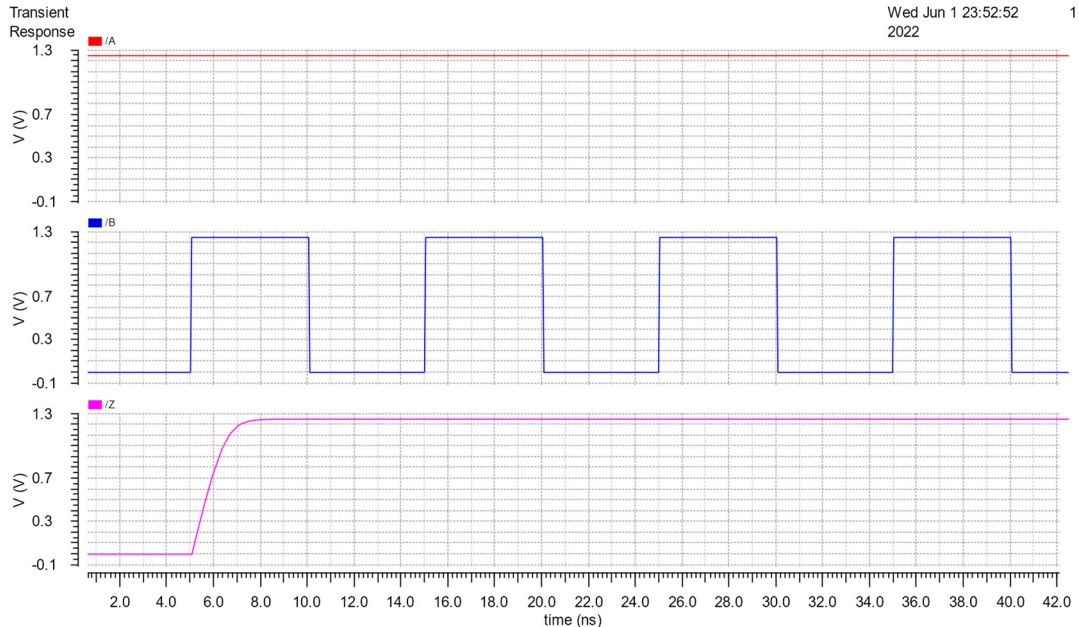

**Figure 10.** Function test results of th22 with A connected V$_{DD}$ and B supplied Vpulse.

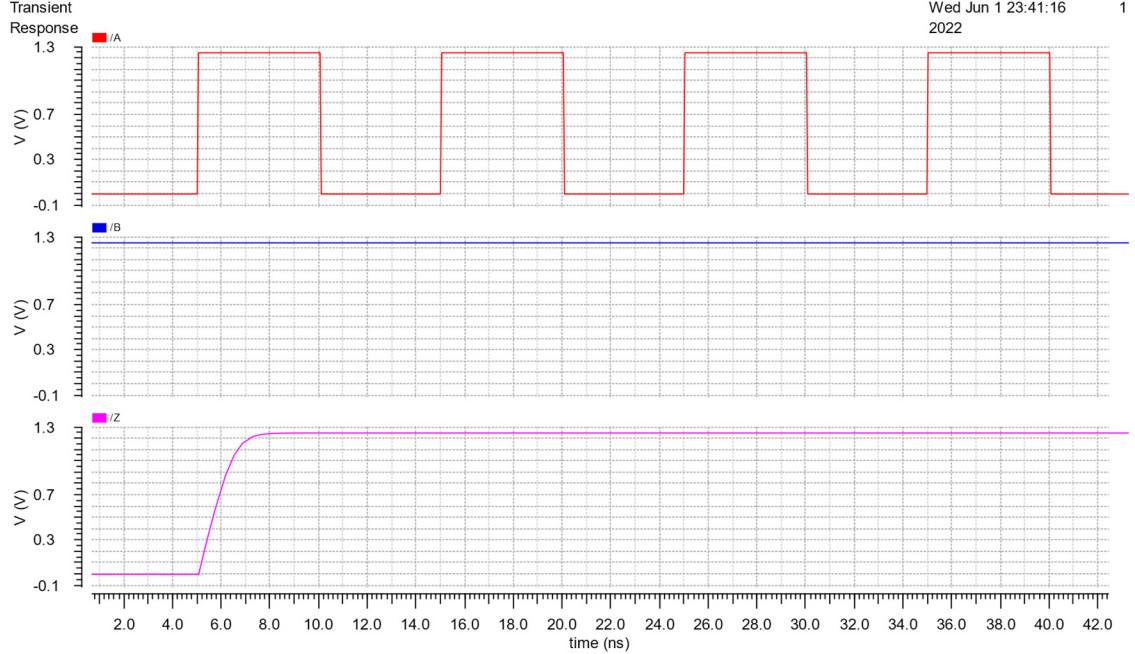

**Figure 11.** Function test results of th22 with A supplied Vpulse and B connected V$_{DD}$.

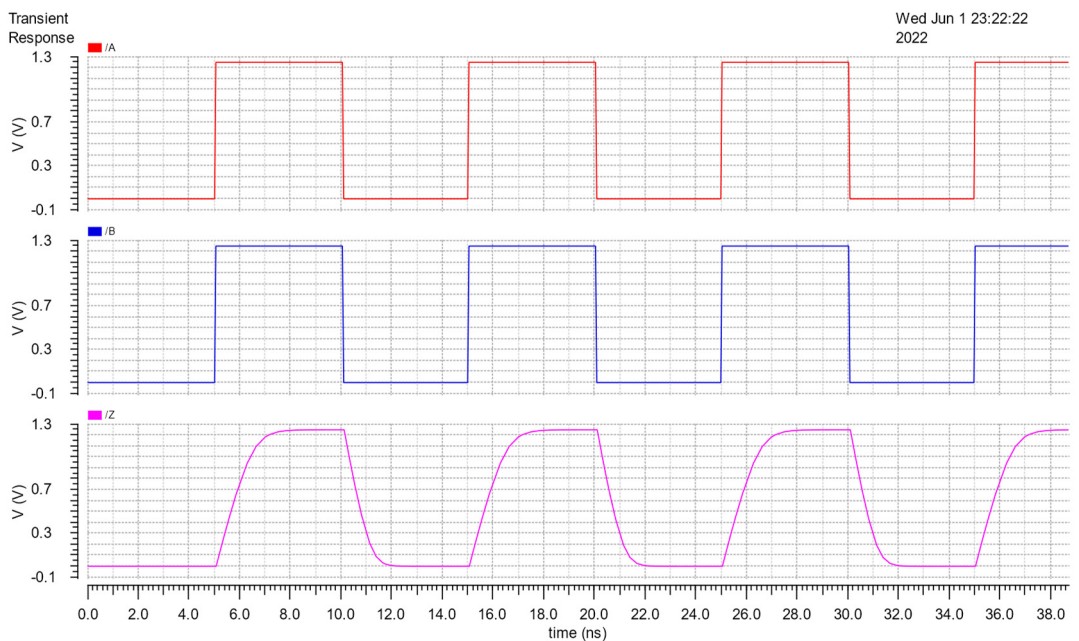

**Figure 12.** Function test results of th22 with A and B supplied Vpulse.

### 3.2. Cell Characterization Results

To implement cell characterization, Ocean script is used to assist in measuring 49 cases as mentioned in Section 2.3. Figure 13 is the simulation result of those 49 cases (with Pin A supplied Vpulse, pin B connected to $V_{DD}$). Similarly, Figure 14 shows the simulation results for the case (with Pin A supplied Vpulse, pin B connected to GND). With the support of Ocean script, the parameters, such as cell fall, cell rise, rise transition, fall transition, rise power and fall power are implemented quickly and accurately. These parameters are shown in Tables 2–7, where the unit of timing parameters is in ns and the power is in pW.

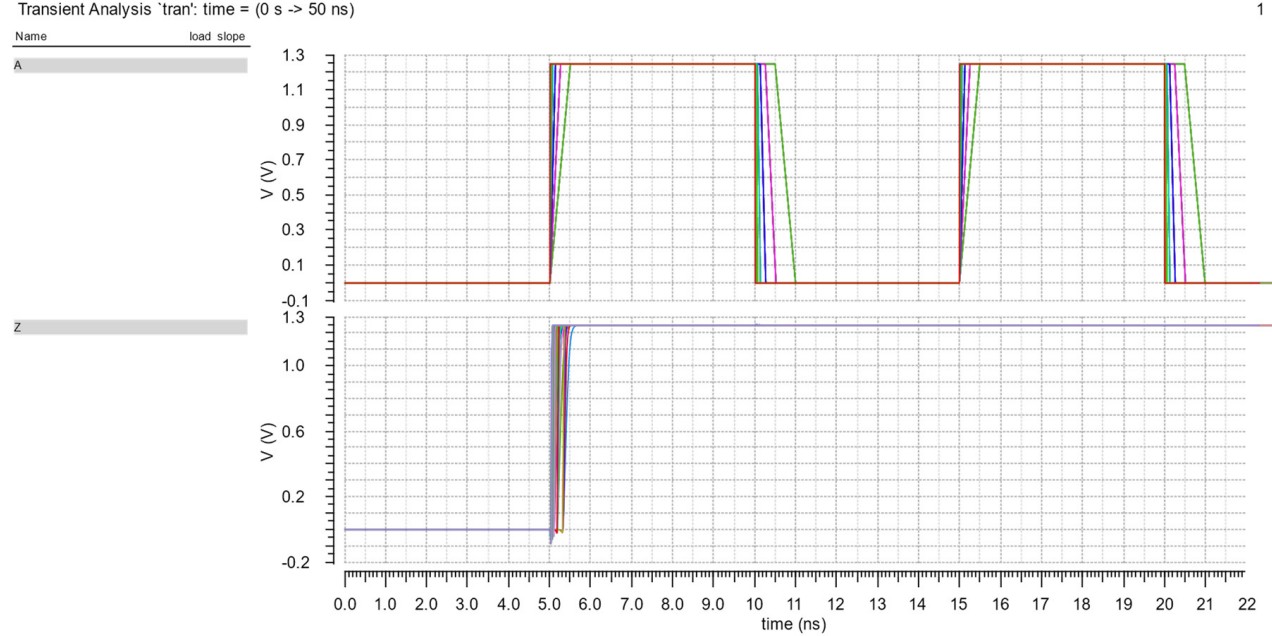

**Figure 13.** The simulation result with Pin A supplied Vpulse and pin B connected to $V_{DD}$.

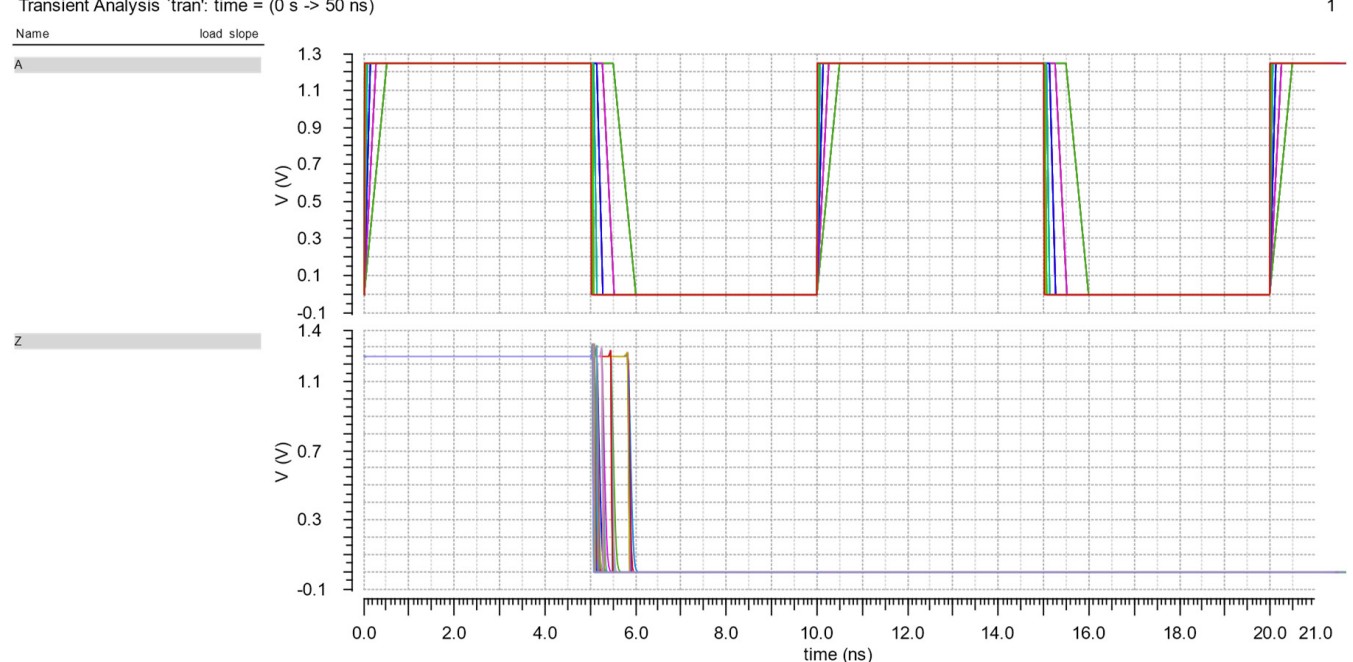

**Figure 14.** The simulation result with Pin A supplied Vpulse and pin B connected to GND.

**Table 2.** Cell rise delay.

| C (fF) | | | | | | |
|:---:|:---:|:---:|:---:|:---:|:---:|:---:|
| **1.4** | **2.54** | **4.61** | **8.37** | **15.2** | **27.6** | **50.0** |
| **T (ns)** | | | | | | |
| 0.0100 | 0.031719 | 0.034713 | 0.039622 | 0.048466 | 0.063802 | 0.091584 | 0.142036 |
| 0.0192 | 0.035519 | 0.038487 | 0.043428 | 0.052211 | 0.067709 | 0.095355 | 0.145774 |
| 0.0368 | 0.042939 | 0.045776 | 0.050817 | 0.059452 | 0.074892 | 0.102274 | 0.153379 |
| 0.0707 | 0.056999 | 0.059810 | 0.064762 | 0.073485 | 0.089004 | 0.117078 | 0.167156 |
| 0.1360 | 0.080912 | 0.083845 | 0.088798 | 0.097410 | 0.112959 | 0.140649 | 0.191456 |
| 0.2610 | 0.120889 | 0.123863 | 0.128831 | 0.137477 | 0.152992 | 0.180382 | 0.229623 |
| 0.5000 | 0.187969 | 0.191297 | 0.196570 | 0.205400 | 0.220554 | 0.248453 | 0.298070 |

**Table 3.** Rise transition.

| C (fF) | | | | | | |
|:---:|:---:|:---:|:---:|:---:|:---:|:---:|
| **1.4** | **2.54** | **4.61** | **8.37** | **15.2** | **27.6** | **50.0** |
| **T (ns)** | | | | | | |
| 0.0100 | 0.014191 | 0.017611 | 0.023507 | 0.034664 | 0.055245 | 0.093438 | 0.162108 |
| 0.0192 | 0.014231 | 0.017607 | 0.023582 | 0.034607 | 0.054907 | 0.092951 | 0.162408 |
| 0.0368 | 0.014317 | 0.017439 | 0.023722 | 0.034508 | 0.055121 | 0.093275 | 0.160903 |
| 0.0707 | 0.014551 | 0.017747 | 0.023710 | 0.035047 | 0.055584 | 0.092349 | 0.162379 |
| 0.1360 | 0.015606 | 0.018867 | 0.024525 | 0.035743 | 0.056056 | 0.093671 | 0.160805 |
| 0.2610 | 0.017896 | 0.021076 | 0.026662 | 0.037100 | 0.056643 | 0.093562 | 0.160940 |
| 0.5000 | 0.021648 | 0.024589 | 0.029991 | 0.039966 | 0.059128 | 0.094819 | 0.160774 |

**Table 4.** Cell fall delay.

| C (fF) | | | | | | | |
|---|---|---|---|---|---|---|---|
| T (ns) | 1.4 | 2.54 | 4.61 | 8.37 | 15.2 | 27.6 | 50.0 |
| 0.0100 | 0.040239 | 0.042706 | 0.046741 | 0.053411 | 0.064899 | 0.084751 | 0.120116 |
| 0.0192 | 0.043811 | 0.046269 | 0.050280 | 0.057040 | 0.068285 | 0.088151 | 0.123954 |
| 0.0368 | 0.050510 | 0.052918 | 0.056919 | 0.063691 | 0.074968 | 0.094818 | 0.130357 |
| 0.0707 | 0.063376 | 0.065755 | 0.069815 | 0.076515 | 0.087776 | 0.107599 | 0.142960 |
| 0.1360 | 0.086937 | 0.089327 | 0.093424 | 0.100122 | 0.111618 | 0.131466 | 0.167641 |
| 0.2610 | 0.127368 | 0.129957 | 0.134214 | 0.141081 | 0.152620 | 0.172618 | 0.208160 |
| 0.5000 | 0.196405 | 0.199268 | 0.203878 | 0.211073 | 0.222948 | 0.243133 | 0.279049 |

**Table 5.** Fall transition.

| C (fF) | | | | | | | |
|---|---|---|---|---|---|---|---|
| T (ns) | 1.4 | 2.54 | 4.61 | 8.37 | 15.2 | 27.6 | 50.0 |
| 0.0100 | 0.012919 | 0.015210 | 0.019491 | 0.027197 | 0.041450 | 0.067562 | 0.115309 |
| 0.0192 | 0.012907 | 0.015205 | 0.019663 | 0.027368 | 0.041394 | 0.067405 | 0.114845 |
| 0.0368 | 0.012835 | 0.015253 | 0.019609 | 0.027360 | 0.041525 | 0.067584 | 0.115338 |
| 0.0707 | 0.012921 | 0.015414 | 0.019568 | 0.027387 | 0.041563 | 0.067599 | 0.115411 |
| 0.1360 | 0.013734 | 0.016063 | 0.020303 | 0.028008 | 0.042103 | 0.067605 | 0.115320 |
| 0.2610 | 0.015538 | 0.017841 | 0.022038 | 0.029759 | 0.043457 | 0.068085 | 0.115567 |
| 0.5000 | 0.018421 | 0.021066 | 0.025241 | 0.032553 | 0.046080 | 0.070092 | 0.116487 |

**Table 6.** Fall power.

| C (fF) | | | | | | | |
|---|---|---|---|---|---|---|---|
| T (ns) | 1.4 | 2.54 | 4.61 | 8.37 | 15.2 | 27.6 | 50.0 |
| 0.0100 | −0.000976 | −0.000986 | −0.000996 | −0.001010 | −0.001025 | −0.001038 | −0.001044 |
| 0.0192 | −0.000949 | −0.000962 | −0.000975 | −0.000991 | −0.001005 | −0.001017 | −0.001025 |
| 0.0368 | −0.000922 | −0.000932 | −0.000954 | −0.000969 | −0.000985 | −0.001001 | −0.001010 |
| 0.0707 | −0.000905 | −0.000919 | −0.000920 | −0.000946 | −0.000949 | −0.000961 | −0.000989 |
| 0.1360 | −0.000902 | −0.000907 | −0.000912 | −0.000926 | −0.000945 | −0.000965 | −0.000967 |
| 0.2610 | −0.000889 | −0.000889 | −0.000898 | −0.000910 | −0.000927 | −0.000946 | −0.000961 |
| 0.5000 | −0.000905 | −0.000909 | −0.000916 | −0.000926 | −0.000940 | −0.000958 | −0.000976 |

**Table 7.** Rise power.

| C (fF) | | | | | | | |
|---|---|---|---|---|---|---|---|
| T (ns) | 1.4 | 2.54 | 4.61 | 8.37 | 15.2 | 27.6 | 50.0 |
| 0.0100 | −0.000926 | −0.001105 | −0.001442 | −0.002045 | −0.003127 | −0.005081 | −0.008588 |
| 0.0192 | −0.000909 | −0.001097 | −0.001433 | −0.002036 | −0.003118 | −0.005068 | −0.008578 |
| 0.0368 | −0.000900 | −0.001086 | −0.001420 | −0.002022 | −0.003105 | −0.005057 | −0.008568 |
| 0.0707 | −0.000885 | −0.001072 | −0.001405 | −0.002007 | −0.003090 | −0.005045 | −0.008558 |
| 0.1360 | −0.000892 | −0.001078 | −0.001408 | −0.002001 | −0.003088 | −0.005047 | −0.008559 |
| 0.2610 | −0.000902 | −0.001083 | −0.001405 | −0.002012 | −0.003096 | −0.005039 | −0.008560 |
| 0.5000 | −0.000950 | −0.001131 | −0.001459 | −0.002059 | −0.003141 | −0.005093 | −0.008610 |

At the end of Section 3.2, Monte Carlo simulations under mismatch variations of cell rise, cell fall, rise transition, fall transition, rise power and fall power are shown in Figures 15–20. These simulations are implemented with a 50pF load and 500ps slope. The simulation results are good because of the similarity to the Gauss distribution with a standard deviation of ±3 sigma.

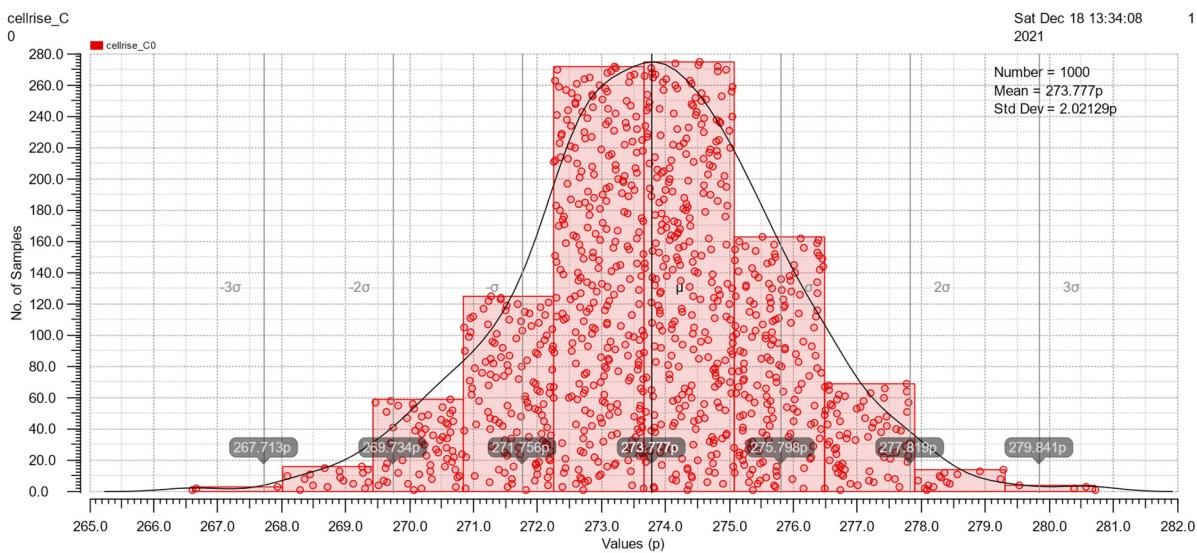

**Figure 15.** The Monte Carlo simulation of cell rise.

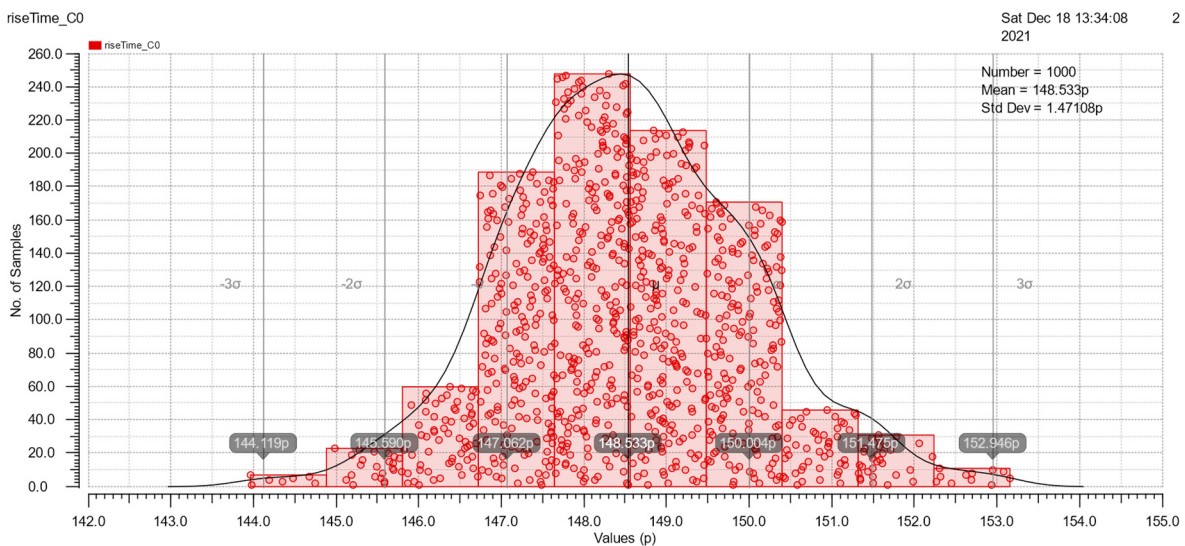

**Figure 16.** The Monte Carlo simulation of rise transition.

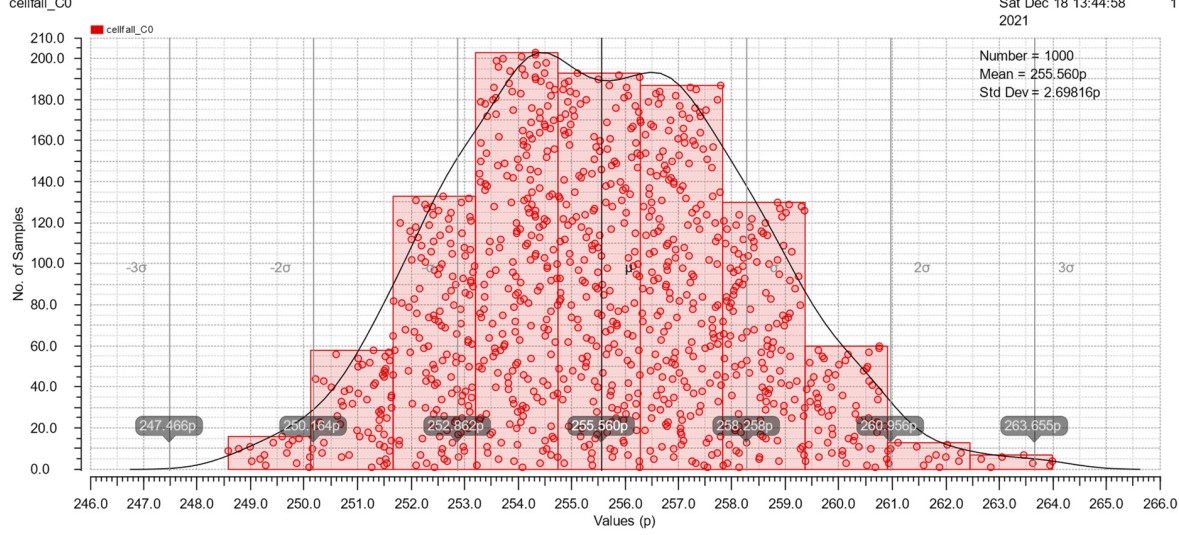

**Figure 17.** The Monte Carlo simulation of cell fall.

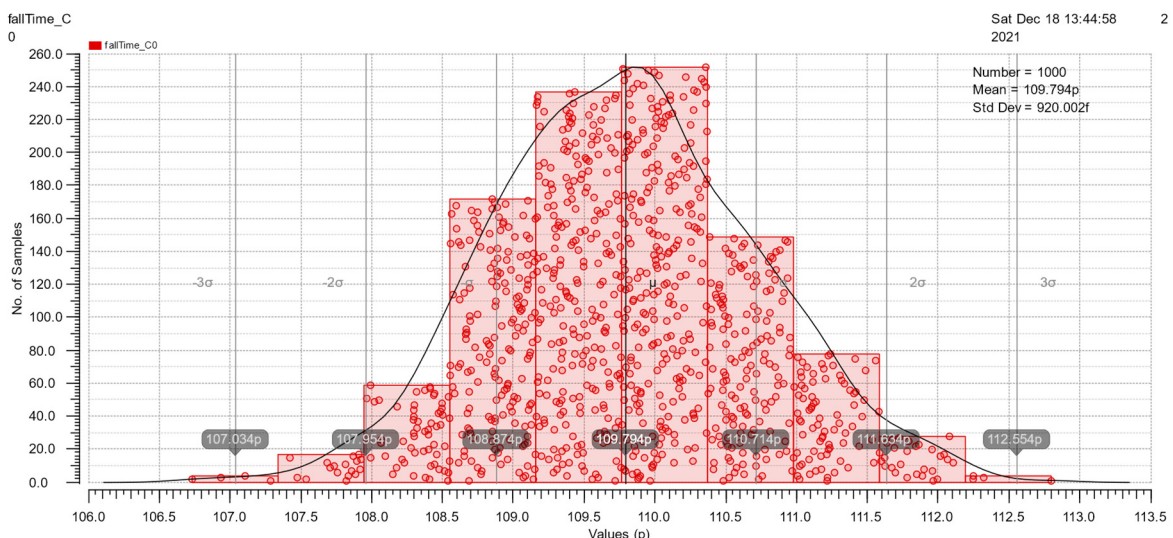

**Figure 18.** The Monte Carlo simulation of fall transition.

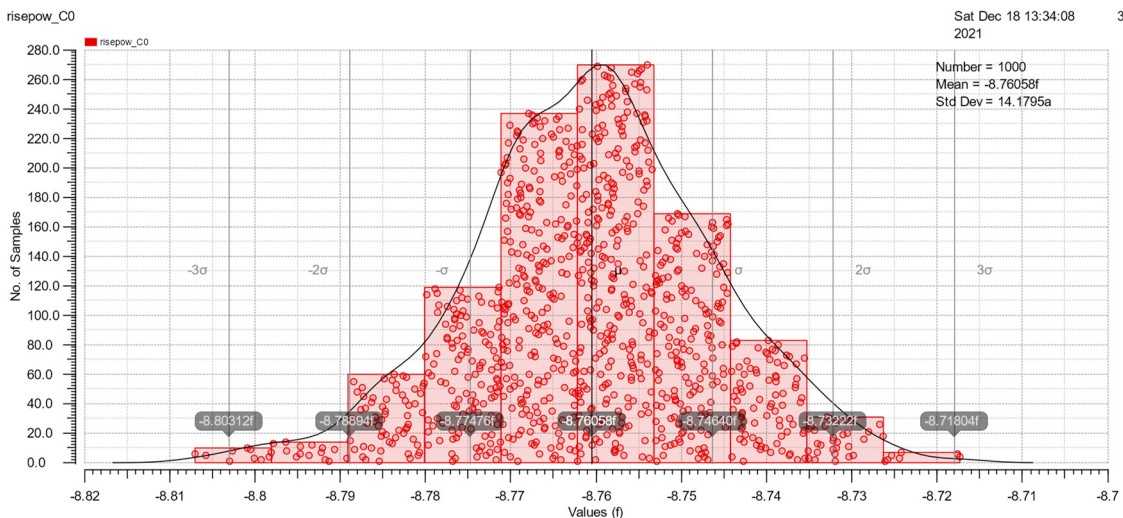

**Figure 19.** The Monte Carlo simulation of rise power.

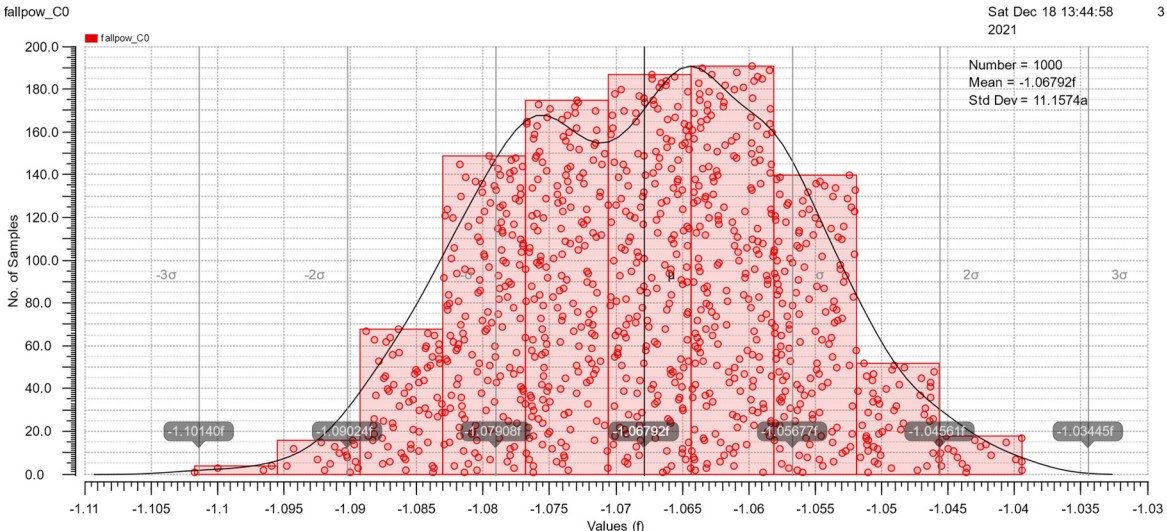

**Figure 20.** The Monte Carlo simulation of fall power.

### 3.3. The Synthesis Results of The RTL Code

In this section, we use the full adder model [26] as an example for testing the library that we generated based on our proposed flow. This model comprises two th23 gates and two th34w2 gates, as shown in Figure 21, and its output equations are as follows:

$$C_{out}^1 = A^1B^1 + A^1C_{in}^1 + B^1Cin^1 \tag{8}$$

$$C_{out}^0 = A^0B^0 + A^0C_{in}^0 + B^0Cin^0 \tag{9}$$

$$\begin{aligned} S^1 &= A^0B^0C_{in}^1 + A^0B^1C_{in}^0 + A^1B^0C_{in}^0 + A^1B^1C_{in}^1 \\ &= C_{out}^0A^1 + C_{out}^0B^1 + C_{out}^0C_{in}^1 + A^1B^1C_{in}^1 \end{aligned} \tag{10}$$

$$\begin{aligned} S^0 &= A^0B^0C_{in}^0 + A^0B^1C_{in}^1 + A^1B^0C_{in}^1 + A^1B^1C_{in}^0 \\ &= C_{out}^1A^0 + C_{out}^1B^0 + C_{out}^1C_{in}^0 + A^0B^0C_{in}^0 \end{aligned} \tag{11}$$

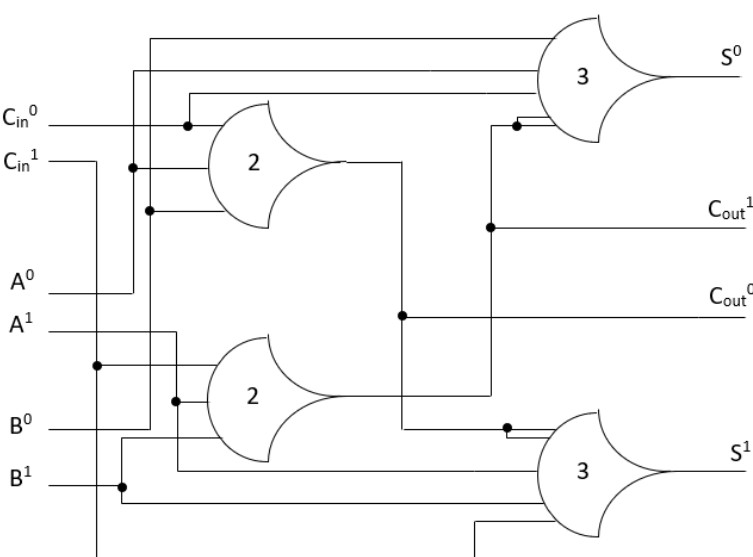

**Figure 21.** NCL full adder.

Since $C_{out}$ is not input-complete with any inputs, S must be input-complete with all inputs [17] which means the equations for S must be in canonical form. Equations (10) and (11) show that the S output has all the inputs (A, B, and $C_{in}$) in each product term. Therefore, the full adder satisfies the input-complete condition.

In Figure 21, the two $Th_{23}$ gates are connected to the two $Th_{34w2}$ output gates, and the outputs for the $Th_{23}$ gates are $C_{out}^1$ and $C_{out}^0$. The $C_{out}^1$ includes the product terms of the inputs (A, B, $C_{in}$) with rail$^1$, and $C_{out}^0$ includes the product terms of the inputs (A, B, $C_{in}$) with rail$^0$. When inputs are asserted, the $Th_{23}$ gates are asserted, and the $C_{out}$ is asserted. Similar to the $C_{out}$ output, $S^0$ includes the product terms of the inputs ($A^0$, $B^0$, $C_{in}^0$ and $C_{out}^1$), and $S^1$ includes the product terms of the inputs ($A^1$, $B^1$, $C_{in}^1$ and $C_{out}^0$). If $C_{out}$ output is asserted, and only one of the three inputs is asserted, the S output will be asserted. For instance, if the inputs (A, B, $C_{in}$) are DATA0, DATA1, and DATA0, respectively, the $Th_{23}$ gates are asserted. As a result, $C_{out}$ and S are DATA0 and DATA1, respectively. Therefore, the circuit shown in Figure 21 satisfies the observability conditions.

We synthesize this by using the Design Compiler. The typical parameters of the library are temperature (25 °C), voltage (1.25 V), and process (ff). The netlist file after synthesis is shown in Figure 22.

```
 1 /////////////////////////////////////////////////////////////
 2 // Created by: Synopsys DC Ultra(TM) in wire load mode
 3 // Version   : L-2016.03-SP1
 4 // Date      : Fri Oct 29 06:55:19 2021
 5 /////////////////////////////////////////////////////////////
 6
 7
 8 module test ( a0, a1, b0, b1, cin0, cin1, rst, s0, s1, cout0, cout1 );
 9   input a0, a1, b0, b1, cin0, cin1, rst;
10   output s0, s1, cout0, cout1;
11
12
13   NCL2W111X1 U5 ( .A(a1), .B(b1), .C(cin1), .Y(cout1) );
14   NCL3W2111X1 U6 ( .A(cout1), .D(a0), .C(cin0), .B(b0), .Y(s0) );
15   NCL2W111X1 U7 ( .A(a0), .B(cin0), .C(b0), .Y(cout0) );
16   NCL3W2111X1 U8 ( .A(cout0), .D(a1), .C(b1), .B(cin1), .Y(s1) );
17 endmodule
```

**Figure 22.** The netlist file after synthesis.

The synthesis results of area, power and delay depicted in Figures 23–25, respectively, show that the NCL-based design is synthesized successfully. Based on our proposed solution, many other cells can be made to create a full set of NCL cell libraries. This work has a substantial contribution to researching and developing the asynchronous circuits based on NCL.

```
Number of ports:                            11
Number of nets:                             10
Number of cells:                             4
Number of combinational cells:               4
Number of sequential cells:                  0
Number of macros/black boxes:                0
Number of buf/inv:                           0
Number of references:                        2

Combinational area:              92.000000
Buf/Inv area:                     0.000000
Noncombinational area:            0.000000
Macro/Black Box area:             0.000000
Net Interconnect area:       undefined  (No wire load specified)

Total cell area:                 92.000000
Total area:              undefined
1
```

**Figure 23.** The area report result.

```
Global Operating Voltage = 1.25
Power-specific unit information :
    Voltage Units = 1V
    Capacitance Units = 1.000000pf
    Time Units = 1ns
    Dynamic Power Units = 1mW      (derived from V,C,T units)
    Leakage Power Units = 1pW

  Cell Internal Power  =    5.4994 uW   (89%)
   Net Switching Power  = 677.1222 nW   (11%)
                          ---------
Total Dynamic Power   =    6.1766 uW   (100%)

Cell Leakage Power    =    5.2171 nW
```

**Figure 24.** The power report result.

```
Timing Path Group (none)
----------------------------------
Levels of Logic:             2.00
Critical Path Length:        0.13
Critical Path Slack:         uninit
Critical Path Clk Period:      n/a
Total Negative Slack:        0.00
No. of Violating Paths:      0.00
Worst Hold Violation:        0.00
Total Hold Violation:        0.00
No. of Hold Violations:      0.00
----------------------------------
```

**Figure 25.** The delay report result.

The comparison between our work and [20] is given in Table 8. In terms of area of the designs, the full adders in [20] are much less than our result because the adders P-FA-L0, P-FA-L1, and P-FA-L2 are strong indication adders that use the common split-end reset and hysteresis mechanism at the circuit level instead of designing each rail separately [20]. These adders share transistors between rails in three configurations, such as logic block 0 (LB0), logic block 1 (LB1), and logic block 2 (LB2), which results in P-FA-L0, P-FA-L1, and P-FA-L2 models. The area of the adder P-FA-L2 is the smallest because it shares the transistors among four rails. As a result, short paths between VDD and GND through the dP transistors at the DATA state are formed [20]. However, reducing more areas makes short paths not static and consumes high power. That is why the P-FA-L2 adder's power consumption is the highest, approximately 1.28 times our result. The power of P-FA-L0 and P-FA-L1 is lower than ours because transistors are shared between the rails. Thus, with the delay, there is a significant difference between the results in [20] and our result because we calculated the delay based on the Design Compiler tool that helps optimize the design while the adders in [20] were measured by using the Cadence tool. Another reason would be due to the influence of the technology node; we used the 45nm technology in our work while the adders in [20] were simulated with the 65nm technology. Therefore, the comparison of delay would only be relative.

**Table 8.** 1-bit full adder comparison results (without registers).

| Design | Area (transistor) | Power (μW) | Delay (ns) |
|---|---|---|---|
| Ours | 92 | 6.17 | 0.13 |
| P-FA-L0 [20] | 74 | 3.57 | 137.44 |
| P-FA-L1 [20] | 66 | 3.77 | 137.9 |
| P-FA-L2 [20] | 60 | 7.93 | 138.66 |

Finally, our static NCL library is compared with the static NCL library in [15]. We notice that both works use the static structure of the NCL cells. The library in [15] was implemented using the author's own tools and the commercial tools. Hence, if there are any problems during the installation and the use, it would be difficult for readers to overcome. Meanwhile, our static NCL library was implemented by commercial tools. The flow to implement this library would also be simpler than that in [15]. In addition, we synthesized the 4-bit full adder by using our NCL library and the NCL library in [15]. The results synthesized by the DC tool are shown in Table 9. In terms of power, the synthesis result using our library is smaller than that using the library in [15]. The reason for the difference in power could be that our library was implemented in the pre-layout stage and the library in [15] was implemented in the post-layout stage. In addition, the synthesis result of delay using our library is larger than the one using the library in [15] because the library in [15] was optimized by many of the author's own tools and implemented in the post-layout stage.

**Table 9.** 4-bit full adder comparison results with two different libraries.

| Design | Power (mW) | Delay (ns) |
|---|---|---|
| Ours | 0.1245 | 1.13 |
| Using library in [15] | 0.1571 | 0.59 |

## 4. Conclusions

In this paper, the methodology to design the NCL cell library was presented via the proposed flow. All blocks of this flow were explained in detail and some examples were given. Our proposed flow could be used for research at universities. It not only could solve the problem of the lack of a standard NCL cell library that is difficult for students and researchers, but also it could help them save time and effort. The complete cell library includes 27 cells which were designed using 45 nm CMOS technology and were used for the synthesis of the NCL-based asynchronous designs by the Design Compiler tool from Synopsys.

**Author Contributions:** Conceptualization, methodology, T.L.T. and T.H.; software, data curation, L.T.T.; investigation, T.L.T. and L.T.T.; writing—original draft preparation, T.L.T.; writing—review and editing, supervision, T.H. All authors have read and agreed to the published version of the manuscript.

**Funding:** This research received no external funding.

**Data Availability Statement:** The data presented in this study are available on request from the corresponding author.

**Acknowledgments:** We acknowledge the support of time and facilities from Ho Chi Minh City University of Technology (HCMUT), VNU-HCM for this study.

**Conflicts of Interest:** The authors declare no conflict of interest.

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
