# Peer review of "A Methodology to Design Static NCL Libraries"

_jlpea, doi:10.3390/jlpea12020031_

Round 1
Reviewer 1 Report
I would to really appreciate the authors for bringing up the improved manuscript with the necessary modifications according to previous comments.
After doing thorough review, I am recommending this paper towards the acceptance in this journal.
Author Response
Thanks for your comments.
Reviewer 2 Report
Comments to authors are in the attached PDF

Author Response
Thanks for your comments. I have uploaded the file. Please see the attachment.

Reviewer 3 Report
The paper presents a methodology for design of NCL libraries.The authors suggest that state-of-the-art NCL library design flows are complex and heavily rely on in-house tools. As a solution they propose a flow that only uses conventional tools, which is claimed to be particularly helpful for researchers. The usability of the proposed flow is demonstrated by creating and characterising one of 27 gates in 45nm technology none. Basic adder circuits are used as benchmarks for comparing the newly created NCL library against a couple of previous works.
My main concern is that parts of the paper are weakly connected and do not form a complete story. There is not enough details in the paper to reproduce the proposed methodology. Several generic equations and a flow chart are not sufficient for the reader to recreate a library of NCL gates. The choice of characterisation parameters (e.g. the values for Cload and Vpulse in section 2.3) are not explained.
Design of NCL gate in the proposed methodology is largely a manual effort - the user is expected to capture transistor-level schematic and validate each gate before proceeding to the characterisation step. This cumbersome task should be significantly simplified via automatic generation of transistor-level implementation (possibly with appropriate transistor sizing) and a simulation testbench for a given NCL gate. While NCL gate characterisation is partially automated via Ocean scripts, these, however, are not made available to the reader.
Comparison against the previous works is rather shallow and does not explain the observations - this needs a lot more work.
The design in [23] uses 180nm technology node, which cannot be directly compared to the proposed 45nm NCL library. The authors' explanation that the difference in delay and power is due to Design Compiler optimisations is questionable. There is not much optimisation possible for a 1-bit adder to justify 20x delay and 200x power difference.
When doing comparison to ASCEnD-FreePDK45 in [15], the 4x difference for delay of 4-bit adder must be properly studied and explained. There is also some inconsistency in delay and power figures for 4-bit adder (e.g. delay is reported as 1.13ns in Table 9, and then as 2.61ns in Table 11).
Author Response

(The authors gave the same response as above.)

Round 2
Reviewer 2 Report
Authors have improved the paper in accordance with reviewers' suggestions, the work can be published, congratulations.
Author Response
Thanks for your valuable comments.
Reviewer 3 Report
The revised version of the paper slightly improved the "Materials and Methods" section adding more details on the design flow for characterisation of NCL gates. There is also a new Table for 1-bit adder and a correction of an error in the "Results and Discussion" section. This, however, is not a major review of the paper. Comparison against the previous works is still rather shallow and does not explain the observations - this needs a lot more work. Note that designs in different technology nodes (e.g. 45nm and 180nm) cannot be directly compared for delay and power. Every significant difference in the obtained results against the previous work must be properly studied and explained.
Author Response

(The authors gave the same response as above.)

Round 3
Reviewer 3 Report
Thank you for the effort and recognising the misleading comparison.
I still believe the comparison against the previous works is shallow and does not attempt to explain the observations. More work is needed here, but the revision timeframe does not allow it. Therefore I leave the decision about acceptance to the editors.
Author Response
Thanks for your comments. I have uploaded the files. Please see the attachment.

This manuscript is a resubmission of an earlier submission. The following is a list of the peer review reports and author responses from that submission.
Round 1
Reviewer 1 Report
I would like to appreciate the authors for introducing a standard semi static NCL cell library for asynchronous designs. But I found some technical confusions in the manuscript.
- In line 124, the authors wrote "the proposed flow depicted in Figure 6 " ,but stated as " Figure 6. Standard NCL Cell Library Design Flow chart ",for figure 6 diagram description which is completely confusing, please clarify.
- The authors need to present their proposed library flow chart separately as another figure with necessary explanation for it.
- There are NCL libraries developed before, so the authors need to compare their proposed cell library with the existing libraries in terms of technology and other parameters.
- Then the authors need to present a comparison table with all the existing libraries including the proposed library.
- It would be more interesting to the readers if the authors includes summary in the results and discussion section about that comparison table.
- More explanation is needed for .db file conversion from*.lib file. Since the authors explained till .lib file generation only.
- The authors need to compare their proposed work with the recent works from 2018,2019,2020 also along with the compared one from 2017.
The authors are expected to modify the manuscript in according with the comments above mentioned.
Reviewer 2 Report
In this paper a design methodology to synthesize semi-static NCL libraries which could be employed in QDI logic has been proposed.
The paper deals with simulation results of the library cells which are then used to successfully synthesize a NCL-based FA.
The Topic of NCL-based circuits and design approach is relevant, but it is not clear the novelty of the approach. Furthermore, some sections are difficult to read; for example at row 114-118 authors affirm that the hold function of semi-static cells can be neglected for specific applications, but no-reference are provided and it is not clear the context, it would be much more effective to provide references for specific applications, or at least to give a more detailed discussion.
At row 293 authors affirm that the area consumption could be improved with respect to ref [18], for example which technique could be used to improve the layout area? In addition, authors should improve the comparison with the state of the art in order to make the work more valuable. Furthermore Fig. 9-10-11 are blurred and moreover it is evident that these Fig.s are screenshot (Fig.11 still contains the label of the screenshot). Authors should redraw these figures and uniform labels with the text. Also figures 12 (a,b,c) are very-difficult to read. Authors should elaborate the *.csv files extracted from Cadence, so as figures would be more readable. Additionally, colours of signals in figures are misleading, signals would be much more readable if the colours are consistent from figure to figure.
Moreover one of the main drawbacks of circuits based on threshold symmetries are that under Mismatch or Process, Voltage and Temperature (PVT) variations the performance would drastically be reduced. It is clear that the proposed FA can work in typical condition but, what about Corners and PVT variations? Authors should report Montecarlo Simulation under Mismatch variations and also PVT variations and cross corners for figures of merits reported in Tab3-11. On top of that, most concerns rely on results presented. It is not clear the novelty of the approach, authors should stressed the aspect of the novelty with respect to other works.